# Development of Gamma Oscillation during Sentence Processing in Early Adolescence: Insights into the Maturation of Semantic Processing

**DOI:** 10.3390/brainsci13121639

**Published:** 2023-11-26

**Authors:** Mohammad Hossein Behboudi, Stephanie Castro, Prasanth Chalamalasetty, Mandy J. Maguire

**Affiliations:** 1School of Behavioral and Brain Sciences, The University of Texas at Dallas, Richardson, TX 75080, USA; behboudi@utdallas.edu (M.H.B.);; 2Callier Center for Communication Disorders, The University of Texas at Dallas, Dallas, TX 75235, USA; 3Department of Human Development and Family Sciences, The University of Texas at Austin, Austin, TX 78705, USA

**Keywords:** gamma, theta, language development, EEG, semantic

## Abstract

Children’s ability to retrieve word meanings and incorporate them into sentences, along with the neural structures that support these skills, continues to evolve throughout adolescence. Theta (4–8 Hz) activity that corresponds to word retrieval in children decreases in power and becomes more localized with age. This bottom-up word retrieval is often paired with changes in gamma (31–70 Hz), which are thought to reflect semantic unification in adults. Here, we studied gamma engagement during sentence processing using EEG time–frequency in children (ages 8–15) to unravel the developmental trajectory of the gamma network during sentence processing. Children heavily rely on semantic integration for sentence comprehension, but as they mature, semantic and syntactic processing units become distinct and localized. We observed a similar developmental shift in gamma oscillation around age 11, with younger groups (8–9 and 10–11) exhibiting broadly distributed gamma activity with higher amplitudes, while older groups (12–13 and 14–15) exhibited smaller and more localized gamma activity, especially over the left central and posterior regions. We interpret these findings as support for the argument that younger children rely more heavily on semantic processes for sentence comprehension than older children. And like adults, semantic processing in children is associated with gamma activity.

## 1. Introduction

Efficiently retrieving and integrating word meanings into a coherent sentence is a complex process in which children perform quickly and seemingly effortlessly. Nevertheless, children continue to enhance their language skills through middle childhood and adolescence [1,2,3,4,5,6,7,8]. As children progress through their school years, their ability to retrieve semantic information becomes faster and more efficient [7,9]. Their ability to integrate this information to comprehend a sentence changes with age as well [6,10,11,12]. Recent research using time–frequency analysis of EEG in adults has shown that changes in neural oscillations, particularly in theta (4–8 Hz) and gamma (31–70 Hz), are associated with retrieving and integrating semantic information, respectively [5,13,14,15,16,17,18]. Despite these findings, there remains a dearth of research investigating the neural oscillatory activity that supports sentence processing in children. A handful of studies have shown developmental differences in theta response during sentence processing in children [2,19]; however, further research is needed to fully understand the complex neural mechanisms that children use as they retrieve words and integrate them into coherent sentences.

In adults, gamma oscillations are strongly associated with the process of semantic unification during sentence processing [14,18,20,21]. Specifically, semantically correct sentences generate a significantly greater gamma power increase than word lists and sentences with semantic violations, even when the syntax remains intact, and this effect intensifies further when semantic integration becomes more challenging [14,18,20,21]. Furthermore, there is a link between gamma activity and prediction in sentence processing, suggesting that gamma oscillations may be involved in predicting upcoming linguistic input [15,22,23]. Beyond language, gamma has been implicated in other areas of processing including motor preparation [24,25,26], visual processing [27], memory encoding [28,29,30], and memory maintenance [31,32,33,34]. Here, we will focus on gamma’s role in sentence processing.

### Related Work 

Neurodevelopmental studies offer invaluable insights into the maturation of cognitive functions, revealing that while basic cognitive processes are established by childhood, the sophisticated coordination of these abilities continues to develop throughout adolescence [35,36,37]. This ongoing development can be attributed to brain maturation and the child’s experiences [36,37,38,39]. By early childhood, aspects such as cortical folding, brain size, and regional functional specialization of the brain largely achieve a form similar to that of adults. However, significant enhancements, including the expansion of dendritic arborization, synaptic pruning, and increased myelination, continue well into adolescence [37]. 

Broca’s area and Wernicke’s area, two key language-related regions in the human brain, in both adults and children are studied in [40]. The findings suggest that the connection between these two regions, which is facilitated by the arcuate fasciculus/superior longitudinal fasciculus (AF/SLF), is less mature in children, potentially due to lower myelination. In children, an alternative pathway to Wernick’s area via the ventral pathway is observed [41]. The temporal lobe matures uniquely, with early maturation of the temporal poles and late maturation of the posterior superior temporal gyrus [40,42]. This area integrates functions of other relatively developed association areas, such as memory, audiovisual association, and object recognition [43]. Aside from neural structure development, neural oscillation development enhances our understanding of language maturation.

In children’s sentence processing, similar to adults’, theta is associated with word retrieval as it appears in response to word presentation [2,44]. However, theta activity in children exhibits a distinct topographic and temporal pattern [5,19,44]. Specifically, in children aged 8–11 years, theta activity is larger in amplitude and is broadly distributed, while in adolescents, it tends to localize to the left central-posterior areas, and the temporal pattern is temporally prolonged compared to adults [2,44,45]. These findings suggest that although theta activity is involved in language processing at an early age, memory systems that involve semantic retrieval and unification may take longer to fully develop.

While our understanding of children’s theta oscillations in sentence processing is growing, the role of gamma oscillations is still relatively unexplored. Resting-state or baseline gamma activity is associated with language processing and vocabulary [46,47,48]. Children with language-learning impairments who take part in intervention exhibit changes in their resting-state gamma activity [49]. Differences in resting-state gamma also mediate the relationship between socioeconomic status and language abilities [47]. However, understanding the dynamics of gamma oscillations during language comprehension tasks in children will offer valuable insights into the involvement of gamma oscillations in language networks. 

Behavioral, fMRI, and ERP studies have provided important insights about how children’s semantic processing develops. Around age 5, children rely heavily on semantic integration for sentence comprehension, but as they mature, semantic and syntactic processing units become distinct and localized, improving into early adulthood [3,4,6,50]. The early reliance on semantic processing during sentence comprehension is thought to be due to the ventral pathway, which responds to the semantic context, serving as a primary connection within the language network during early language development [42,51,52,53,54,55]. In adults, syntactic processing is mainly observed in the superior temporal gyrus (STG) and pars opercularis [4,6,52,53,56,57]. However, in children, semantics and syntax units overlap and engage additional areas of the left and right Inferior Frontal Gyrus (IFG) [4,6,8,58].

This study aims to investigate the differences in gamma engagement between middle childhood and adolescence during sentence processing. By using time–frequency analysis, we will provide new insights into how gamma activity, which corresponds to semantic integration in adults, is engaged to support sentence comprehension between middle childhood and early adolescence. We use the same data set as in our previous study on theta engagement [2], but expand it to gamma. In the previous work, we found that theta power increases and then decreases quickly as children are exposed to each word in the sentence. We predict the temporal pattern of gamma activity will be different showing a longer period of heightened activity spanning across words to support the process of semantic unification as the sentence unfolds [14,18,59]. Across age, we expect to see this increase in gamma amplitude becomes smaller and more localized [3,4,6,50]. This would be interpreted as evidence that younger children heavily rely on semantic integration for sentence comprehension, whereas as they age, their reliance on this semantic unification decreases and the associated neural networks become more specialized and localized. 

## 2. Materials and Methods

### 2.1. Participants

Ninety-eight children, spanning ages 8 to 15 years, were included in this analysis. This sample size surpasses those of previous developmental and adult language studies allowing for robust statistical analysis [5,10,14,44]. This group was selected from a larger study of 150 participants, and those with noisy EEG signals were excluded. Participants were grouped by age: 8–9 years (Mean = 9.15, SD = 0.44, N = 20; 6 female), 10–11 years (Mean = 11.04, SD = 0.47, N = 28; 18 female), 12–13 years (Mean = 13.02, SD = 0.61, N = 24; 13 female), 14–15 years (Mean = 14.77, SD = 0.55, N = 26; 14 female). All of the participants included in the analysis were right-handed and, based on self-reports, had no history of significant neurological issues, no language development delay or disorders, and no reading disabilities at the time of the study. Furthermore, all the participants were English-dominant. Following a brief introductory period, the consent documents were completed by the parents, and the assent documents were completed by the children.

### 2.2. Linguistic Stimuli

Each participant was presented with 100 sets of sentence triplets, wherein each sentence ended with a pseudoword that served as a noun and was preceded by an article or a possessive pronoun (See Table 1). The sentence length varied between 6 and 9 words wherein the words included were selected from the MacArthur-Bates Communicative Inventories, a corpus of words spoken by children between the ages of 4 and 5 years. The sentence structures vary such that the location and frequency of verbs are different across sentences. The verb occurs between words one and seven, but most commonly is the third or fourth (mean = 3.38, See Table 2 for some examples). Each word appeared for 500 milliseconds (ms) and a 300 ms blank screen appeared between each word. The participant was then asked what word best fit all the sentences in the triplet. In our analysis, we focused on the first five words of each sentence to study neural oscillation, ensuring that we examined the same amount of data across all sentences.

### 2.3. EEG Acquisition

To collect EEG data, each participant was asked to sit on a chair 1 m away from the monitor that was used to present stimuli. They were told that they will see three sentences, word by word, and at the end of each triplet, they needed to guess what word fits the last word for all three sentences. EEG data were acquired with a 64-electrode Neuroscan cap, in which the electrodes were placed in the standard 10-10 system. EEG data were recorded continuously using a Neuroscan SynAmps2 amplifier and CURRY software (V7, https://compumedicsneuroscan.com/curry-epilepsy-evaluation/, (accessed on 25 November 2023)) sampled at 1 kHz with impedances typically below 5 kΩ. Data were recorded with the ground at Fz and the reference electrode located near the vertex.

### 2.4. EEG Pre-Processing

To pre-process EEG signals, data were re-sampled to 250 Hz. Then, a zero-phase FIR filter using a hamming window was used to filter the data with a low-pass filter at 80 Hz, and high-pass filter at 0.1 Hz. The noise line (60 Hz) was removed using a notch filter with a 2 Hz transition bandwidth. This filter, which is zero phase and non-causal, has cutoff frequencies at −6 dB from 59 to 61 Hz. All bad electrodes were removed using the “clean_rawdata” EEGLab plug-in to detect signals beyond the normal amplitude, which were removed by considering +/−75 µv as the threshold [60,61]. To remove non-cortical signals, we used ICA decomposition and the MARA plugin, a machine-learning artifact rejection algorithm [62,63], in the EEGLab toolbox [61]. MARA is a robust, supervised method designed to identify and reject various types of artifacts, including but not limited to eye movement, heartbeats and muscle movement artifacts, thereby offering a comprehensive solution for artifact rejection [62,63]. After eliminating artifacts and bad channels, the data were referenced to the average potential across all the electrodes. Then, the data were epoched from 500 ms before the onset of the sentences to 3700 ms after (end of word 5). Then, the baseline of the data was removed by considering 350 ms before the onset of the first word as the baseline. 

### 2.5. Time–Frequency Analysis

Time–frequency analysis was used in our study to compute event-related spectral perturbations (ERSP). All eligible participants entered the study design and were grouped into four different conditions based on age (8–9 years, 10–11 years, 12–13 years, and 14–15 years). For ERSP, the calculation was based on the Morlet wavelet with the wavelet cycle beginning from 2 cycles and increasing linearly by 0.5 cycles in order to have more cycles for higher frequencies. This resulted in the wavelet width ranging from 2 cycles at 3 Hz to 23.70 cycles at 71 Hz. The EEGLab toolbox was used to calculate the ERSPs between 3 and 71 Hz for each electrode and across each sentence (trial). The average power for each trial across all electrodes and frequencies was calculated and normalized by subtracting the baseline mean for that specific electrode and frequency. This allows us to track changes in brain activity over time and across different frequencies. Then, to obtain the topographic map of gamma (31 to 70 Hz) activity, we average the activity across the time and frequency over the scalp to see the spatial distribution of the activity over the scalp. 

### 2.6. Statistical Analysis

After calculating all the ERSPs for each group, a between-subject analysis of variance (ANOVA) was conducted to compare the neural activity (dependent variable: ERSP in each Time × Frequency × Electrode points) between different age groups (independent variable: age groups: 8–9 years, 10–11 years, 12–13 years, and 14–15 years). The non-parametric permutation-based ANOVA was calculated using EEGLab with a *p*-value of 0.05 to find the main effect of age on ERSP. To correct for multiple comparison hypothesis testing, False Discovery Rate (FDR) correction was used. FDR is a technique used in multiple hypothesis testing to account for the anticipated percentage of false positive outcomes among the rejected null hypotheses (Type I Error) [64,65]. In order to keep the expected rate of false discoveries among all made discoveries under control at a specified level (here 5%), it modifies the *p*-value threshold used to declare statistical significance. This is helpful when several tests are run since it lowers the possibility of creating a lot of false positive findings and improves the specificity of the outcomes. For further analysis, Tukey’s HSD post-hoc analysis was conducted. This ANOVA was run on both theta and gamma frequencies to determine if there was a significant difference in the activity pattern of these frequencies across ages. The findings related to the theta frequency band have been previously documented in our earlier publication [2]. In the current manuscript, our primary focus is on the gamma frequency band. However, we make references to the theta analysis where relevant for a comprehensive understanding. Region of interest (ROI) analysis was included, as opposed to reporting findings from each individual electrode, to quantify the ERSP findings better and to simplify and streamline the ANOVA results. Electrodes were categorized into 6 regions of interest and the average ERSP in each ROI was calculated for further statistical analysis. The six ROIs are Left Frontal (FP1, AF3, F7, F5, F3, F1); Left Central (FC5, FC3, C5, C3, C1, CP5, CP3, CP1); Left Posterior (P7, P5, P3, P1, PO7, PO5, PO3, O1); Right Frontal (FP2, AF4, F2, F4, F6, F8); Right Central (FC2, FC4, FC6, C2, C4, C6, CP2, CP4, CP6); Right Posterior (P2, P4, P6, P8, PO4, PO6, PO8, O2) electrodes.

## 3. Results

### 3.1. Spatial Analysis

To understand the development of gamma activity during sentence processing, we performed an omnibus one-way ANOVA (Figure 1), which revealed significant differences between age groups in the topography of gamma engagement averaging over the course of sentences (0–3700 ms). Specifically, younger children display higher gamma amplitudes over the scalp, but for older children, this effect is much smaller and occurs only over the left frontocentral and parietal regions. The main effect of age is statistically significant (*F*(3, 94) = 4.08, *p* = 0.009; *η_p_*^2^ = 0.12) over the left frontal region as well as the left central region (*F*(3, 94) = 3.35, *p* = 0.022; *η_p_*^2^ = 0.10), right frontal (*F*(3, 94) = 4.23, *p* = 0.007; *η_p_*^2^ = 0.12), and right central region (*F*(3, 94) = 3.91, *p* = 0.011; *η_p_*^2^ = 0.11). The difference between gamma power over the right and left posterior regions is not significantly different among this age group.

The results of the Tukey post-hoc analysis revealed statistically significant differences in various electrode measurements across different age group pairs (Figure 2). Specifically, children aged 8–9 years exhibited significant differences from those aged 12–13 years in the left frontal and right fronto-central electrodes. Additionally, they also differed significantly from the 14–15 years old group in the right fronto-central and central electrodes. On the other hand, children in the 10–11 years age group showed significant differences from the 12–13 years old group in the left frontal, central, and centroparietal electrodes. Furthermore, this age group also exhibited significant differences from the 14–15 years old group in the central electrodes. These findings highlight the distinct variations in electrode measurements across different age groups.

### 3.2. Spatiotemporal Analysis

The spatial analysis revealed the main effect of age on gamma oscillation across the scalp. By studying changes in gamma activity over the course of the sentence, with a specific focus on when the words are displayed and when they disappear (Figure 3), we can observe how gamma networks act at each stage of sentence processing and as the sentence unfolds, As mentioned in the previous section (Section 3.1). An omnibus one-way ANOVA was conducted at each step of sentence processing to identify the specific stages that contribute to the observed significant differences in gamma oscillation among the different age groups. The results show no obvious differences between the gamma response to a word and to a blank between words, unlike what was observed in theta activity [2]. Instead, this effect changes over the course of the sentence, peaking around words 3 and 4 (1600 to 2900 ms). Our analyses revealed no age group differences at word 1, but after word 2, the gamma response becomes larger for younger age groups, eventually resulting in a significant age group differences (around word 3, 1600 ms after the onset of the first word and word 4, 2400 ms after the onset of the first word). This effect is largest over the right hemisphere for word 3 and becomes more widely distributed by word 4 before decreasing throughout the remainder of the sentence.

A detailed examination of the ANOVA was conducted using Tukey’s HSD post-hoc analysis, focusing on pairwise comparisons between different age groups as the sentence unfolds. The analysis revealed statistically significant differences in gamma power during sentence processing between younger children (8–11 years old) and older children (12–15 years old), as illustrated in Figure 4. Notably, the most pronounced differences were observed between children aged 10–11 years and those aged 12–13 years, with variations extending across multiple electrodes. Also, the largest amount of significantly different electrodes occurs between word 3 and word 4 (1600–2900 ms). 

### 3.3. Temporal Analysis

The temporal analysis was conducted to elucidate the temporal pattern of gamma activity during sentence processing as each word is presented and as sentence unfolds. The average ERSP across the gamma frequency band (31–70 Hz) was calculated, focusing on the electrodes that showed significant results in the ANOVA (Figure 1). These selected electrodes included FP1, F7, F5, F1, F8, FC5, FC3, FCz, FC2, FC6, FT8, C5, C3, C1, Cz, C2, C4, T8, CP5, CPz, TP8, CP6, P7, and P6. The results are presented in Figure 5. The overall pattern from the temporal analysis indicates that gamma activity increases with the presentation of each word with a delay relative to the theta activity observed in [2]. This increase in gamma activity is more prolonged in children aged 8–11 years. For instance, the gamma increase for the first word begins at approximately t = 400 ms, and for children aged 8–11 years, it continues even after the presentation of the second word (around t = 1000 ms after the onset of the sentence). However, in children aged 12–15 years, the increase ceases around 800 ms, followed by a decrease in gamma activity. In the 8–11 age group, even after the decrease phase, the gamma activity remains higher than the baseline. This represents the engagement of a neural network in younger children that is even observable during word presentation. Conversely, for ages 12–15, average gamma amplitudes decrease after words 2 and 3 (t = 2400 to 3700 ms), such that by the end of word 5, the gamma amplitude returns to baseline levels. This suggests distinct patterns of neural processing across different age groups during sentence comprehension. The independent sample t-test was conducted on a selection of 20 random participants from each age group, with the aim of identifying statistically significant timepoints across the different age groups. The findings revealed a notable distinction in the temporal pattern of gamma activity between younger children (aged 8–11 years) and older children (aged 12–15 years) with the alpha level of 0.05. Specifically, the results indicated that the gamma activity in younger children (8–11 years old) exhibited a higher amplitude from the onset of the second word (t = 800 ms) in comparison to the older children (12–15 years old). This heightened activity persisted throughout the entire sentence until it unfolded. 

## 4. Discussion

This is the first paper to study how gamma power related to sentence processing changes between middle childhood and early adolescence. We observed that gamma activity increased gradually from the beginning of the sentence (slightly later than theta activity) and remained elevated even between words as the sentence unfolds (Figure 2). This is consistent with adult studies that showed gamma is related to semantic unification, or the prediction of the upcoming word in sentence processing and gamma is present along the sentence, rather than a response to each word [14,15,18,20,21,22,66]. We also observed that gamma shows a developmental shift around age 11, with the younger groups (8–9 and 10–11) exhibiting broadly distributed gamma activity with higher amplitudes and older groups (12–13 and 14–15) exhibiting smaller and more localized gamma activity, especially in the left central and posterior regions (Figure 1). Similar to studies of children’s sentence comprehension, this indicates that children’s reliance on semantic processing during sentence comprehension decreases during middle childhood and adolescence and becomes more localized [6,8,12,50,58].

Gamma and theta have been associated with semantic aspects of sentence processing, with theta reflecting individual lexical retrieval and gamma reflecting integration. Our data support this distinction because the temporal aspects of gamma in our study are notably different from what has been found in the past associated with theta [2,5,45]. Previous studies [2,5,45] observed theta increase in response to individual words in a sentence, which is broadly distributed in younger children and decreases in amplitude and localizes to over the left centro-parietal regions in older children. Further, these bursts in theta activity occur shortly after the onset of each word and largely attenuate between words. As this study utilized data from the previously mentioned study conducted by Maguire et al. 2022 [2], which specifically investigated the effect of theta on sentence processing, our study did not report detailed results regarding theta. Our contribution lies in the integration of gamma activity along with the well-established frequency band in children’s sentence processing, theta. 

Through word-level temporal analysis, we observed an increase in gamma activity, after the theta burst, which typically occurs at the onset of individual words [2,44]. The rise in gamma power suggests the activation of a neural network responsible for integrating new words with their preceding context (Figure 5). The peak of this gamma activity occurred around the third and fourth words, followed by a decrease in activity (Figure 2). Older children exhibited a faster and more drastic decrease in gamma activity, returning to baseline levels by the end of the fifth word. Additionally, we observed a gradual increase in gamma activity up to the point that we anticipate to be the maximum semantic unification load followed by a gradual decrease thereafter. We interpret this finding as support for the argument that gamma activity is associated with semantic unification during sentence processing [14,20,21,28]. While our study does not specify the type of unification, previous adult studies associated gamma with semantic unification rather than syntactic unification [14,20,21,28].

In adults, there is also evidence that gamma is related to the congruency between the expected and observed input or prediction [15,66]. However, our observation of a gradual decrease in gamma power after the third or fourth word does not align with these interpretations as words appearing later in a sentence are often more predictable and therefore easier to process [66,67]. 

The temporal pattern of our gamma results is particularly interesting because it appears to peak around the sentence’s main verb, indicating that verbs are strengthening the semantic integration of sentences. While verbs are not the main focus of this study, previous research has shown the significant role of verbs in the semantic structure of sentences [68,69]. Future studies will need to investigate the relationship between verb phrase processing and gamma activity during development. 

Although our main goal was not related to where in the brain the effect originates, topographic maps can give us a general idea of how the neural engagement changes with age. Topographically, younger children exhibited a broader distribution of gamma activity, whereas in older children, this activity is mostly localized, specifically to electrodes over the left central and posterior regions (Figure 1). These patterns appear to be consistent with findings that syntactic units develop later than semantic units and, as children get older, semantic and syntactic processing units are segregating gradually and become more specific [4,6,8,42]. This is in line with previous work showing that in adults, syntactic processing primarily engages the superior temporal gyrus (STG) and pars opercularis [6,52,53,57]. On the other hand, children showed overlapping brain activity for both semantics and syntax. This supports previous work that indicates children engage additional areas of the left and right inferior frontal gyrus (IFG) [4,6,58,70]. The reliance on semantic processing during sentence comprehension is thought to be related to the ventral pathway being a primary connection within the language network during early language development [42,52,53].

While our study provides valuable insights into gamma-band oscillations during sentence processing in children, it does have certain limitations. Although we observed effects at the main verb in our gamma analysis, our study was not specifically designed to investigate verb location. This is because the sentences were not aligned in such a way that the verb occupied a specific location. Therefore, while we can suggest a correlation, we cannot definitively confirm this effect. Furthermore, our current focus on gamma-band oscillations could be expanded in future research. For instance, a cross-frequency coupling analysis could be employed to explore the relationship between changes in theta oscillations and changes in gamma oscillations. This approach could offer a more comprehensive understanding of the connection between different cognitive functions in sentence processing. Ultimately, it could lead to a more holistic understanding of the oscillatory dynamics underlying sentence processing in children. 

This paper addressed the development of gamma activity during sentence processing tasks from childhood to adolescence. The analysis of gamma frequency during sentence processing in children alongside the previous theta studies advanced our knowledge about the change in children’s semantic unification ability. This paper adds to the small but growing body of literature about the neural oscillatory correlates of language processing and how they continue to change throughout middle childhood and adolescence. In addition to expanding our knowledge about developmental cognitive neuroscience, this work will build an essential foundation for studies focused on how processes might differ in children with language delays and disorders.

## Figures and Tables

**Figure 1 brainsci-13-01639-f001:**
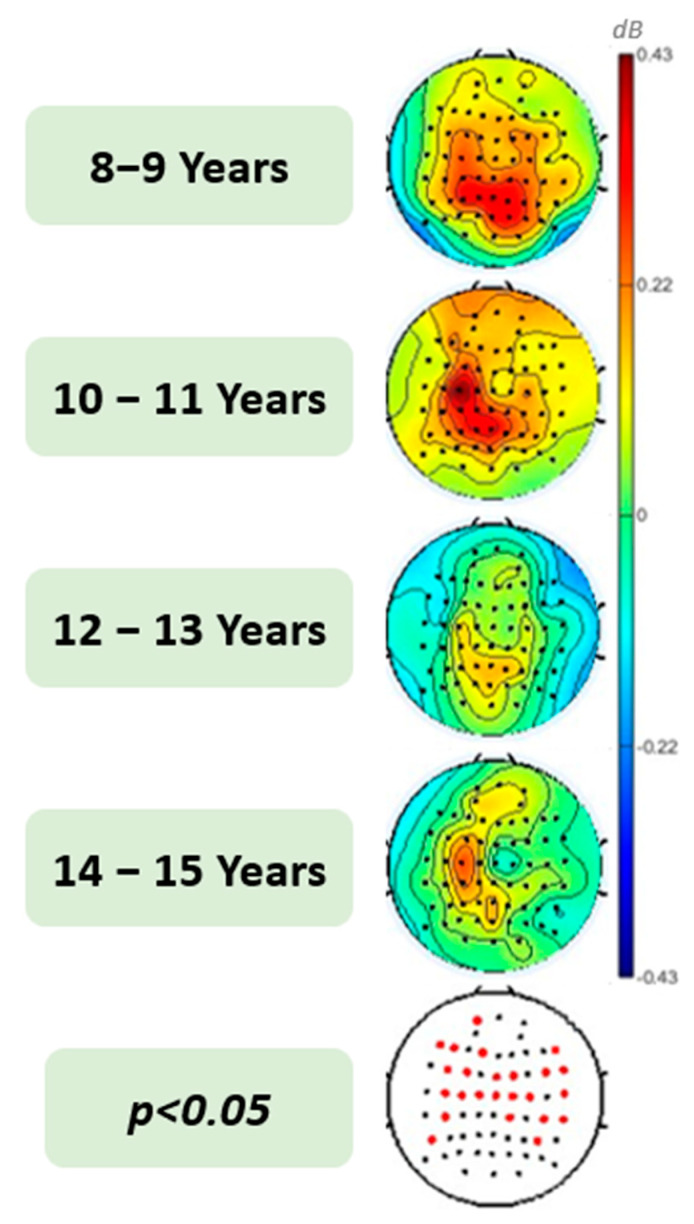
Topographical distribution of average gamma activity across sentences in children aged 8–15 years. The red dots on the *p*-value map signify electrodes that have yielded statistically significant results in the ANOVA test, thereby indicating the main effect of age.

**Figure 2 brainsci-13-01639-f002:**
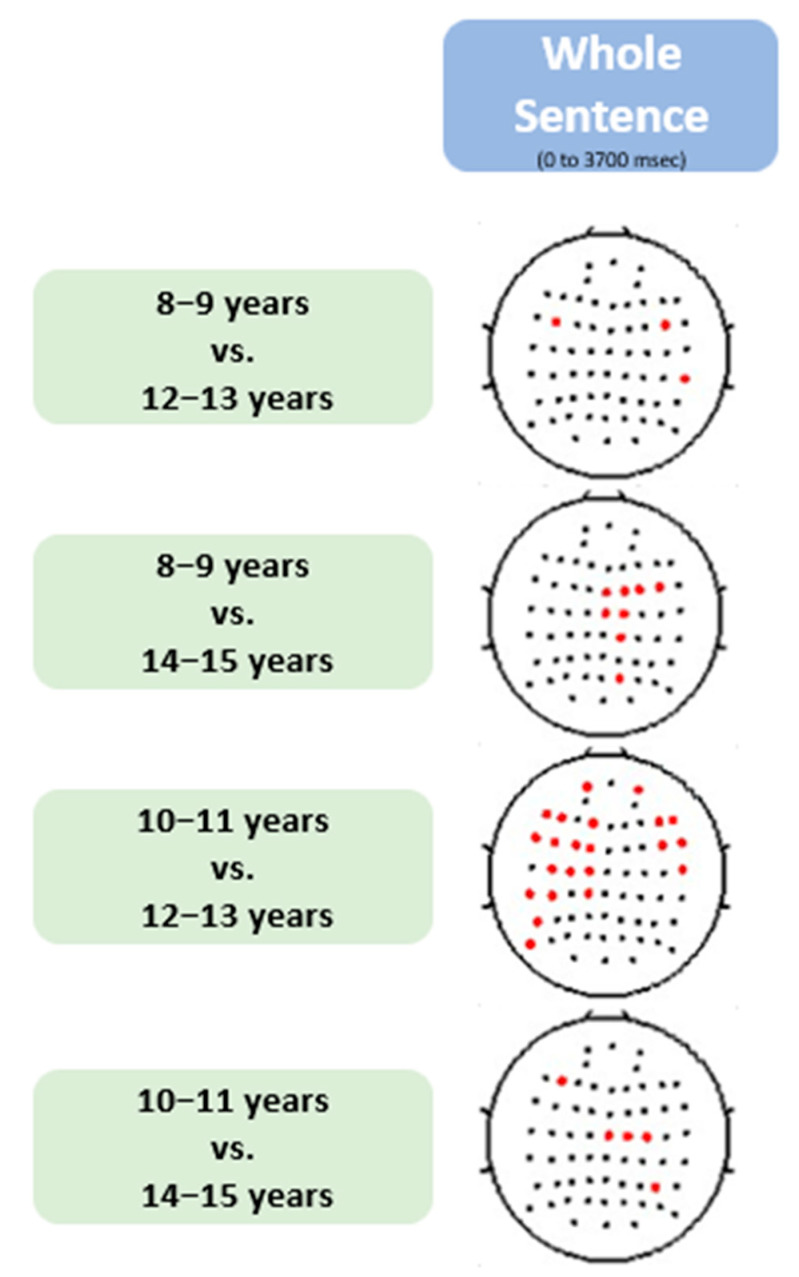
Post-hoc analysis on ANOVA results in Figure 1. The significant electrodes are highlighted in red in each pairwise comparison, with an alpha level set at 0.05. The rows represent *p*-values for pairwise comparisons between different age groups: the first and second rows compare 8–9-year-olds with 12–13 and 14–15-year-olds, respectively, while the third and fourth rows compare 10–11-year-olds with 12–13 and 14–15-year-olds, respectively. Notably, the pairwise comparisons between 8–9 and 10–11year-olds, and between 12–13 and 14–15-year-olds, were not statistically significant.

**Figure 3 brainsci-13-01639-f003:**
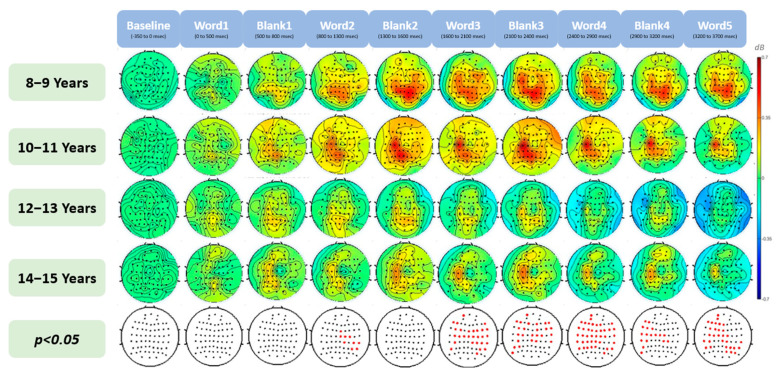
Topographical distribution of gamma band event-related spectral perturbation (ERSP) changes during the course of sentence processing in children aged 8–15 years. The red dots in the *p*-value map indicate the electrodes that show a significant effect of age in the ANOVA analysis amongst all age groups (*p <* 0.05).

**Figure 4 brainsci-13-01639-f004:**
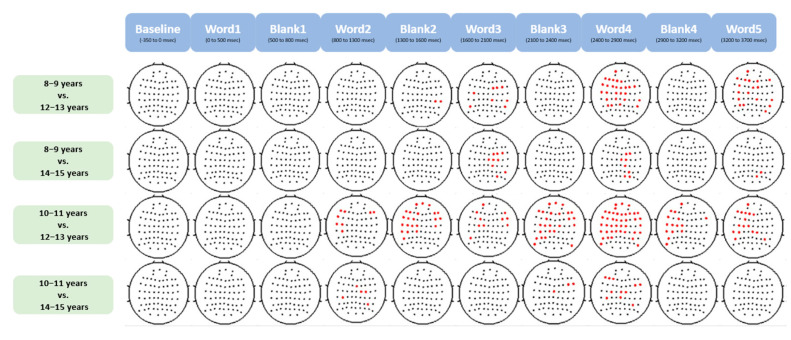
Post-hoc analysis on ANOVA results in Figure 3. The significant electrodes are highlighted in red in each pairwise comparison as sentence unfolds, with an alpha level set at 0.05. The columns show different stages of sentence presentation (i.e., word presentation and blank between words). The rows represent *p*-values for pairwise comparisons between different age groups: the first and second rows compare 8–9-year-olds with 12–13 and 14–15-year-olds, respectively, while the third and fourth rows compare 10–11-year-olds with 12–13 and 14–15-year-olds respectively. Notably, the pairwise comparisons between 8–9 and 10–11-year-olds, and between 12–13 and 14–15-year-olds, were not statistically significant.

**Figure 5 brainsci-13-01639-f005:**
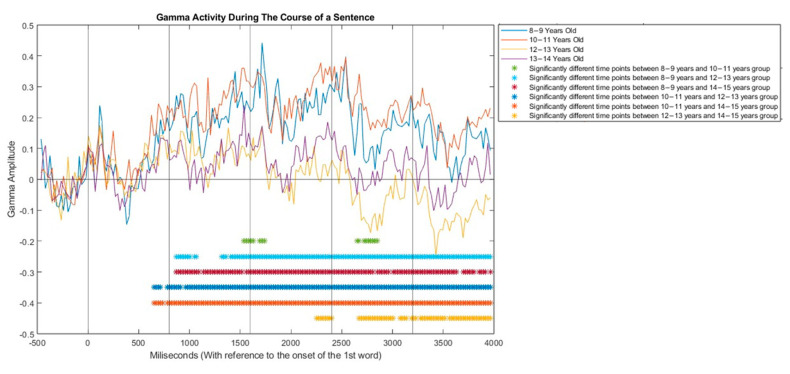
The averages of gamma band power oscillation (in dB) across the electrodes that show significant results in the ANOVA (Figure 1) during the course of sentence processing in children aged 8–15 years. Each vertical line represents the onset of word presentation, and the horizontal line indicates the average of baseline level of gamma power. The starts show represents the time points that are significantly (*p* < 0.05) different in each pairwise comparison.

**Table 1 brainsci-13-01639-t001:** Example of sentence triplets presented to the participant.

Sentence 1	Go outside and play with the neg	Target word: **Cat**
Sentence 2	I’m covered with hair from the neg	
Sentence 3	All night, I heard the meowing of the neg	

**Table 2 brainsci-13-01639-t002:** Verb position in the stimuli.

He **spilled** food on his wesh.
The dog **slept** on the presh.
To keep the sun away, I **wear** a ruch.
Hi favorite toy of all time **is** the therg.
When I **sit** there, I **can** see the pleg.

## Data Availability

The data for this study are available online as part of the Open Science Framework: https://osf.io/z38rb/ (accessed on 25 November 2023).

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
