# Peer review of "Development of Gamma Oscillation during Sentence Processing in Early Adolescence: Insights into the Maturation of Semantic Processing"

_brainsci, 2023, doi:10.3390/brainsci13121639_

Round 1

Reviewer 1 Report

Comments and Suggestions for Authors

Behboudi and collegueas showed the results of an EEG study on children regarding the development of semantic process. For this purpose, they recruited a large cohort of children by dividing it into 4 age groups. The population size is remarkable and the research field interesting.  My main concerns regards the statistical analysis and the presentation of the results.

For the statistical analysis, the authors selected ANOVA tests. Did the authors test the homogeneity of variances before proceeding with the ANOVA? What post-hoc tests did the authors apply to evaluate differences between groups?

I'm confused by the results. I can understand that the results of the spatio-temporal analysis are only described (no statistical data are reported) and I would expose them as the first introductory paragraph to the results, but the results of the spatial and temporal analyzes leave me perplexed.

Spatial analysis. Authors described differences between areas. How did they performed this analysis? How did they defined the areas? Are the areas the electrodes? What is “medium effect size”? Which groups are different? Maybe, as well as a revision of the text, a table with values per groups and statistical results could clarify. For clarity, it would be better to reiterate the temporal boundaries (0- ms?).

Temporal analysis. Here, the presentation of the results is truly lacking. What do the lines in figure 3 that are described in the paragraph represent? The average of all electrodes? Or some areas? Did they perform a statistical analysis? And the results? Why the authors did not perform statistical analysis on the mean value in each temporal windows?

With this lack of statistically significant results, it is not possible to comment on the discussion. I only point out some inaccuracies in a scattered manner: authors referred to cortical areas like IFG, pars opercularis, STG etc etc but they did not perform source localization nor they described accurately the topography of their results (the last paragraph of discussion seems useless, no limitations of the study are mentioned and reference is made to figure 4.

Moreover, I think that the statement on informed consent is incorrect as the participants were children.

Reviewer 2 Report

Comments and Suggestions for Authors

This paper presented an EEG study related to the sentence processing in a 8-15 years old children. The goal was to explore whether gamma band oscillations decrease with age. In general, the paper contains usefull information on a timely topic and is well-written. However, there are some concerns regarding some aspects of the article, especially regarding the methods and results presentation. The comments are listed below:

Abstract: "Temporally, gamma increases as each word is presented, this gamma power remains higher than the baseline between words in children aged 8-11 years, but more readily returns to baseline levels between words in 12-15 year-olds." -> Really difficult to follow this sentence. Please rephrase because this key result is not worded correctly.

Abstract: "We interpret these findings as support for the argument that younger children rely more heavily on semantic processes for sentence comprehension than older children. And like adults, semantic processing in children is associated with gamma activity" -> Why is this central conclusion? Based on what the authors said in the beginning of Abstract, gamma power has been associated with semantic unification. Why this gamma decrement with age means that chlidren rely more heavily on semantic processe for sentence comprehension? Why theta is not included in the analyses?

Introduction: Towards improving readability by a general reader, tt would be beneficial for the Introduction to have a separate "Related Work" section. In this section, the authors can include general-purpose studies about key neurodevelopmental findings in childhood and adolescents, especially focusing on the maturation of some key neuronal circuitry (sensory system, language processing system, memory system, etc). Additionally, the role of gamma/theta oscillations should be also added in the frame of those circuits (what's the role of gamma-theta in languange/memory/sensory systems?). Also, it is worth incuding the following articles in this paragraph, plus any other authors' preference:

1. Luna, B., & Sweeney, J. A. (2001). Studies of brain and cognitive maturation through childhood and adolescence: a strategy for testing neurodevelopmental hypotheses. Schizophrenia bulletin, 27(3), 443-455.

2. Giannopoulos, A. E., Zioga, I., Papageorgiou, P., Pervanidou, P., Makris, G., Chrousos, G. P., ... & Papageorgiou, C. (2022). Evaluating the modulation of the acoustic startle reflex in children and adolescents via vertical EOG and EEG: Sex, age, and behavioral effects. Frontiers in Neuroscience, 16, 798667.

Introduction and Analysis: The authors refer to their previous article on theta-band differences in childhood during sentence processing.

In my opinion, the correct (and more interesting) way to included gamma-band oscillations in this analysis is to perform a cross-frequency coupling study, in order to try to show the extent to which changes in theta produce/correlate with changes in gamma. Given that this analysis requires substantial changes in the article, the authors should at least include this type of analysis as limitation or future direction.

Introduction: The predictions of the authors in the last paragraph are a bit biased. It seems that all the hypotheses are made upon observing

the resultsof the study. I suggest to remake all of them so as to rely on existing knowledge and literature, rather than on the authors' intuition.

Section 2.4. How did you deal with the line noise of 50 Hz which affects the gamma band activity?

Section 2.4: Why did you perform first the average reference? This approach produced artifact leakage to all of the EEG channels, since if a channel is noisy, then the average reference propagates the noisy data of this channel to the rest of the channels. The authors should first remove noisy channels and the average reference should be done immediately before ICA algorithm.

Section 2.4: Which plugins used for cleaning? If you want to avoid describing the preprocessing steps, refer to Giannopoulos, A. E., Zioga, I., Kontoangelos, K., Papageorgiou, P., Kapsali, F., Capsalis, C. N., & Papageorgiou, C. (2022). Deciding on optical illusions: Reduced alpha power in Body Dysmorphic Disorder. Brain Sciences, 12(2), 293.

Section 2.4: How many ICs are rejected on average? Did ICA and MARA focus on only rejecting eye-blinks?

Section 2.6. The authors say that the did ANOVA in both bands (theta and gamma). Where are the results about theta? Do these results already published elsewhere?

Section 2.6. Which are the dimension of ANOVA? Specify the between-subject factors, the within-subjects factor and the dependent variable.

Figure 1: Why we have one p-map? If the authros found a main effect of age group, then they did t-tests amongst all pairs of age groups (I guess).

To which pair of age group the p-value map corresponds? Please right concretely what you did because it is very difficult to follow.

Section 3.1. Did you use ROI-based analysis? Why it is not mentioned in the materials and methods? What is Eta2? Do you mean the partial eta coefficient (η_p)^2? Please improve the symbolisms.

Figure 2: To which comparison the p-map corresponds? 

Comments on the Quality of English Language

The English language of this paper needs a proofread and minor edits.

Reviewer 3 Report

Comments and Suggestions for Authors

This study does not make a significant contribution to the literature. The proposed model lacks novelty, the data rendering needs to be improved, the literature needs to be critically analyzed, and the results can be presented in a better way to clarify the future direction.

Comments on the Quality of English Language

Moderate English language editing is recommended. 

Round 2

Reviewer 2 Report

Comments and Suggestions for Authors

The authors addressed all of my comments.

Reviewer 3 Report

Comments and Suggestions for Authors

Authors have addressed most of my comments. 

Comments on the Quality of English Language

Minor English Language editing is recommended.